

# Liver cancer mortality over six decades in an epidemic area: what we have learned

Jian-Guo Chen[1,2], Jian Zhu[1], Yong-Hui Zhang[1], Yong-Sheng Chen[1], Jian-Hua Lu[1], Yuan-Rong Zhu[1], Hai-Zhen Chen[2], Ai-Guo Shen[2], Gao-Ren Wang[2], John D. Groopman[3] and Thomas W. Kensler[3,4]

[1] Department of Epidemiology, Qidong Liver Cancer Institute / Qidong People's Hospital / Affiliated Qidong Hospital of Nantong University, Qidong, Jiangsu, China
[2] Department of Epidemiology, Affiliated Tumor Hospital of Nantong University, Nantong, Jiangsu, China
[3] Bloomberg School of Public Health, The Johns Hopkins University, Baltimore, MD, United States of America
[4] Public Health Sciences Division, Fred Hutchinson Cancer Research Center, Seattle, WA, United States of America

Corresponding authors
Jian-Guo Chen, chenjg@ntu.edu.cn
Thomas W. Kensler, tkensler@fredhutch.org

## ABSTRACT

**Background and aims:**. Liver cancer is one of the most dominant malignant tumors in the world. The trends of liver cancer mortality over the past six decades have been tracked in the epidemic region of Qidong, China. Using epidemiological tools, we explore the dynamic changes in age-standardized rates to characterize important aspects of liver cancer etiology and prevention.

**Methods**. Mortality data of liver cancer in Qidong from 1958 to 1971 (death retrospective survey) and from 1972 to 2017 (cancer registration) were tabulated for the crude rate (CR), and age-standardized rate and age-birth cohorts. The average annual percentage change was calculated by the Joinpoint Regression Program.

**Results**. The natural death rate during 1958–2017 decreased from 9‰ to 5.4‰ and then increased to 8‰ as the population aged; cancer mortality rates rose continuously from $57/10^5$ to $240/10^5$. Liver cancer mortality increased from $20/10^5$ to $80/10^5$, and then dropped to less than $52/10^5$ in 2017. Liver cancer deaths in 1972–2017 accounted for 30.53% of all cancers, with a CR of $60.48/10^5$, age-standardized rate China (ASRC) of 34.78/105, and ASRW (world) of $45.71/10^5$. Other key features were the CR for males and females of $91.86/10^5$ and $29.92/10^5$, respectively, with a sex ratio of 3.07:1. Period analysis showed that the ASRs for mortality of the age groups under 54 years old had a significant decreasing trend. Importantly, birth cohort analysis showed that the mortality rate of liver cancer in 40–44, 35–39, 30–34, 25–29, 20–24, 15–19 years cohort decreased considerably, but the rates in 70–74, and 75+ increased.

**Conclusions**. The crude mortality rate of liver cancer in Qidong has experienced trends from lower to higher levels, and from continued increase at a high plateau to most recently a gradual decline, and a change greatest in younger people. Many years of comprehensive prevention and intervention measures have influenced the decline of the liver cancer epidemic in this area. The reduction of intake levels of aflatoxin might be one of the most significant factors as evidenced by the dramatic decline of exposure biomarkers in this population during the past three decades.

## INTRODUCTION

Liver cancer is one of the most dominant malignant tumors in the world. According to GLOBOCAN 2018 (*Bray et al., 2018*), it was estimated that the incident cases of liver cancer was 841,080, which accounted for 4.65% out of 18,078,957 malignant tumors in the world every year, ranking sixth in the world (fifth in men and ninth in women). Among 9,555,027 deaths with cancer, 781,631 cases (8.18%) died of liver cancer, ranking fourth in the world (second in men and sixth in women) (*Bray et al., 2018*). In 2017, incident cases of liver cancer in China were about 441,200, and the number of deaths were 395,200 (*Liu et al., 2019*), implying that the proportion of incident cases and death cases of liver cancer in China accounts for 52.46%, and 50.56%, respectively, of the global burden. In particular, liver cancer has among the poorest prognosis of all cancers. For instance, according to data of 291 cancer registries in 61 countries and regions in the CONCORD 3 Study (*Allemani et al., 2018*), the 5-year survival rate of liver cancer was 5%–30% in 2000–2014, and only 10%–19% in Europe and America; while the 5-year survival rates in China in 2006–2008, 2009–2012, 2013–2015 were 10.1%, 9.8% and 12.1%, respectively (*Zeng et al., 2018*). In another research report in 2017, it was estimated that the global disability-adjusted life years (DALYs) was 20.8 million, of which 99% was years of life lost (YLLs) due to early death and 1% was years lived with disability (YLDs) (*Global Burden of Disease Cancer Collaboration et al., 2018*). Thus, liver cancer constitutes a serious disease burden both in the whole world and in China, and the study of liver cancer in China is of great significance for global cancer control.

Since the 1970s in China, a series of systematic studies on the epidemiology, etiology and prevention of liver cancer have been carried out, and great progress has been made. Among them, the prevention and treatment of liver cancer in Qidong, formerly a county and now a city in Jiangsu Province, has been most representative (*Zhu, Chen & Huang, 1980*; *Chen et al., 2019*). This paper summarizes the trend of liver cancer mortality over the past 60 years in Qidong according to a death retrospective survey and, subsequently, a cancer registration report system, and makes an analysis and evaluation on the related challenges and progress in the study of the etiology and prevention of this disease. As such, it demonstrates that the age-standardized trajectories of dominant, often fatal cancers can be attenuated dramatically with better understandings of underlying etiologies and implementation of cancer control programs.

## MATERIALS & METHODS

### Retrospective survey of death

During the period of July to August 1972, a 14-year retrospective survey (1958–1971) of cancer was organized among the 1.03 million residents in Qidong in order to understand the prevalence of liver cancer in this area (*Zhu, Chen & Huang, 1980*). At that time, the investigation teams were composed of cadres of brigade (village) doctors, barefoot doctors (grassroots medical staff), county hospital doctors, and medical team members (from hospitals/research institutes/universities in Shanghai and Jiangsu, Province). The investigation was carried out in the form of an investigational forum, usually with 3–5

"old" peasants and cadres in each brigade (village) (*Chen, 2013*). The information in the questionnaire included address, name, gender, age at/and the date of death, cause of the death and/or the main symptoms (if any), and the diagnosis. For suspected cancer deaths, especially liver cancer, double checking of the cause would be performed with their family members or relatives. The survey showed that most of the cases were diagnosed and treated before death. In 1971, for example, 31.25% of the cases with liver cancer were diagnosed by the Shanghai Cancer Hospital and/or other provincial and municipal hospitals; 50.20% of the cases were diagnosed by the Qidong People's Hospital or the Affiliated Hospitals of Nantong Medical College; 14.17% were diagnosed in district and/or commune (township) hospitals; only 4.38% were not identified in diagnosis units (unknown).

## Cancer registration report

The Qidong Cancer Registry was established in 1972, covering the whole Qidong area and all of the registered population. All malignant tumors, including brain and central nervous system tumors, have been included and registered (*Chen, 2013*). The International Classification of Diseases (ICD) 8th edition (ICD-8) was used for cancer cases before 1979, ICD-9 was used for the years 1979–2000, and ICD-10 was used for coding after 2000. Liver cancer was 155.0–155.9 in ICD-8 and ICD-9, and C22.0–22.9 in ICD-10. All data used in this database have been recoded as C22.0–22.9 according to ICD-10.

In 1990, the Qidong Cancer Registry was admitted to the International Cancer Registration Association (IACR) as a voting member unit. Some of data after 1983 were published in "Cancer Incidence in Five Continents (CI5)" by the International Agency on Cancer Research (IARC) (*Parkin et al., 1997*; *Bray et al., 2020*), or "Annual Report on Cancer of China" by the National Cancer Registry Center of China (*He & Chen, 2017a*; *He & Chen, 2017b*). Most of the liver cancer cases in Qidong were diagnosed by alpha-fetoprotein (AFP) and/or B-ultrasound (85.81%), and were verified by histology (12.36%). Almost all cases (98.17%) were defined by at least one of the three diagnostic criteria; classification by death certificate only (DCO) were only 0.34%.

## Population information

Information regarding the population of Qidong has come from the Residents Household Annual Report of the Qidong Public Security Bureau. The population size in Qidong has exceeded 1 M since the 1970's; and reached a level of 1.12M in 2017. The age structure of the Qidong population in the years after 1972 have come from several sources: sampling survey in 1976 (ps1976), the third (national) census in 1982 (ps1982), the fourth census in 1990 (ps1990), the fifth census in 2000 (ps2000) and the sixth census in 2010 (ps2010). Hence, the population numbers of the age-groups from 1972 to 1976 were extrapolated according to the population structure of ps1976; the population age-group from 1982 to 1990 was extrapolated according to the population structure of ps1982 and ps1990; and the same method was used to extrapolate the population numbers of age-groups by period through the structures of two adjacent censuses.
## Statistical methods

For the period of 1958–1971, the Qidong death retrospective survey data (*Zhu, Chen & Huang, 1980*) were used to calculate the natural death rate, cancer mortality and liver cancer mortality. For the period of 1972–2017, the natural death rate came from the Qidong Vital Registry. The other statistical indicators were calculated based upon the registered data, which included the crude mortality rate (CR), age-specific mortality rate, sex-specific mortality rate, truncated rate of 35–64, cumulative rate 0–74, and cumulative risk; and birth cohort analysis was performed. The age-specific mortality rate was calculated as an average level for each five-year period (following the period compiled in CI5 (*Parkin et al., 1997*; *Bray et al., 2020*). The age-standardized rates were calculated by the China population in 1964 (ASRC) and by the world population in 1960 (ASRW), respectively. The growth rate of the two years (y, y-1) at the end of the period compared with the two years (x, x + 1) at the beginning of the period was evaluated by the percentage change (PC) of the rate (r): $PC_{x-y} = \{[(r_y + r_{y-1}) - (r_x + r_{x+1})]/(r_x + r_{x+1})\} \times 100$ (*Chen et al., 2006*). The Joinpoint Regression Program 4.7.0.0 (2019) was used to estimate the annual percent change (APC), average annual percent change (AAPC) and APC of each age group, allowing for the average increasing/decreasing trends of the rates to be evaluated (*National Cancer Institute, 2019*; *Clegg et al., 2009*).

## RESULTS

### Natural death rate, mortality rates of cancer and liver cancer in Qidong

According to the data from this period of six decades, the natural mortality rates in Qidong residents have experienced progressions from high to low, and then from low to high; the mortality rate between 1958 and 1960 was about $900/10^5$ (9‰), which dropped to its lowest of $540/10^5$ (5.4‰) in the middle and late 1970s, and then slowly increased back to about $900/10^5$ (9‰) at the end of the period (2015–2017). Mortality rates of cancer combined increased slowly from $57/10^5$ in 1958 to $200/10^5$ in 1995 now exceed $300/10^5$. The mortality rate of liver cancer increased from $20/10^5$ in 1958 to about $50/10^5$ in the 1970s, then fluctuated towards about $70/10^5$ in the late 1990s and then at the beginning of this century, moved towards $80/10^5$ in 2008, before decreasing continuously, now approaching $52/10^5$ (Fig. 1).

### Liver cancer mortality, cumulative mortality and truncated rate

Cancer registration data after 1972 were used to evaluate accurately the proportion of liver cancer hazard within total cancer deaths. The average crude mortality rate of liver cancer was $58.86/10^5$, while deaths due to liver cancer accounted for 30.53% (31,261/102,390 total cases), and the truncated mortality rate of 35–64 years old was $104.06/10^5$, the cumulative mortality rate of 0–74 years old was 4.79%, and the cumulative risk of liver cancer was 4.68%. The mortality rates and other evolution indicators over the years for liver cancer in Qidong are shown in Table 1.

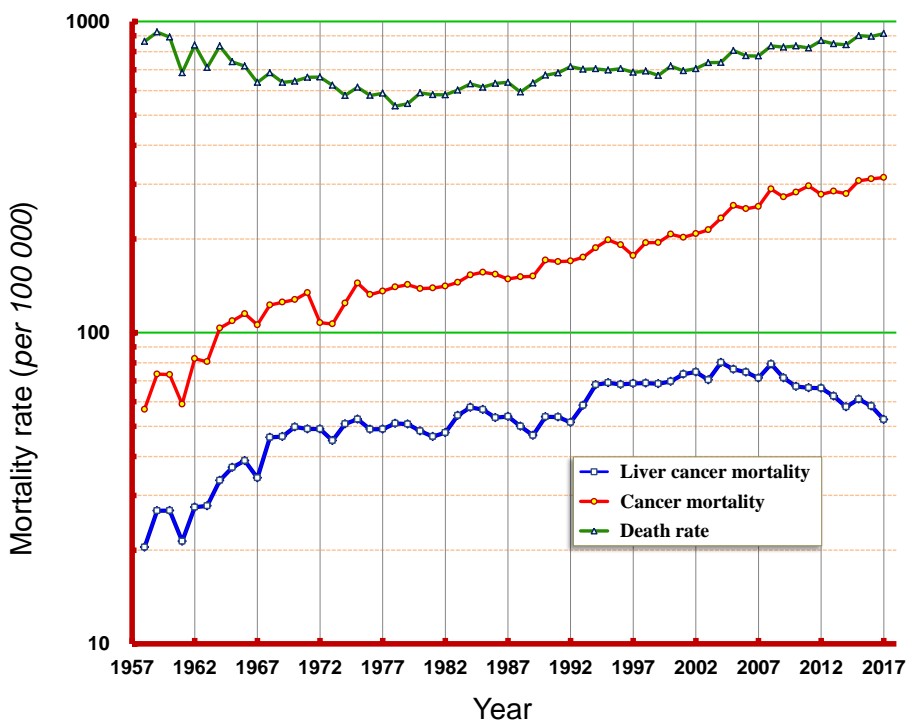

**Figure 1 Natural death rate, mortality rates of cancer and of liver cancer in Qidong, 1958–2017.** Data of 1958–1971 were from a death retrospective survey, while data of 1972–2017 were from cancer registration. Natural death rates in Qidong residents experienced trends from high to low, and to high again; overall cancer mortality rates have shown a continuous increase; whilst liver cancer mortality experienced a dynamic flux from low to high, then sustained at high levels, followed by a gradual decrease in recent years.

## Trend of liver cancer mortality

From 1972 to 2017, the crude mortality rate of liver cancer in Qidong increased by +17.78% in PC, and +0.88% in APC; But, the ASRC and ASRW decreased by −61.25% and −51.43% in PC, and −1.80% and −1.35% in APC, respectively, showing a clear downward trend. Analysis using the Joinpoint Regression Program showed that the average annual percent change (AAPC) of ASRW was −1.3% (95% CI [−1.6% to −1.1%]), in which −1.5% (95% CI [−1.7% to −1.2%]) was for men and −1.0% (95% CI [−1.3% to −0.7%]) was for women, as shown in Fig. 2 and Table 2.

Turning point analysis of multiple models showed that the trend in ASRW of liver cancer during the 45 years of registration in Qidong could be divided into three joinpoints (four periods), i.e., 1972–1992, 1992–1995, 1995–2006 and 2006–2017. Each of the APCs was −1.39%, +4.50%, −1.42% and −4.66%, respectively. For men, it was one point (2 periods): 1972–2005 and 2005–2017, APCs were −0.48% and −4.51%, highlighting a significant downward trend. For women, four joinpoints (5 periods) were identified: 1972–1986, 1986–1989, 1989–1994, 1994–2008 and 2008–2017, APCs were −0.42%, −11.52%, +7.68%, −1.13% and −4.74%, respectively, indicating that the APC of the last period had a significant downward trend (Fig. 3). The results given by the multiple models

**Table 1 Crude mortality, truncated rate and cumulative rate of liver cancer in Qidong,1972–2017.**

| Year | No. of cancer death | No. of liver cancer death | Of total cancer (%) | Liver cancer mortality rate (per $10^5$) | Truncated rate of 35–64 (per $10^5$) | Cumulative rate of 0–74 (%) | Cumulative risk (%) |
|---|---|---|---|---|---|---|---|
| 1958 | 438 | 158 | 36.07 | 20.45 | NA | NA | NA |
| 1959 | 577 | 210 | 36.40 | 26.79 | | | |
| 1960 | 583 | 213 | 36.54 | 26.79 | | | |
| 1961 | 472 | 171 | 36.23 | 21.35 | | | |
| 1962 | 678 | 226 | 33.33 | 27.49 | | | |
| 1963 | 689 | 237 | 34.40 | 27.74 | | | |
| 1964 | 910 | 295 | 32.42 | 33.51 | | | |
| 1965 | 985 | 333 | 33.81 | 36.89 | | | |
| 1966 | 1062 | 358 | 33.71 | 38.77 | | | |
| 1967 | 999 | 322 | 32.23 | 34.13 | | | |
| 1968 | 1183 | 444 | 37.53 | 46.05 | | | |
| 1969 | 1236 | 457 | 36.97 | 46.33 | | | |
| 1970 | 1287 | 502 | 39.01 | 49.80 | | | |
| 1971 | 1377 | 502 | 36.46 | 49.02 | | | |
| 1972 | 1113 | 507 | 45.55 | 49.04 | 138.43 | 5.95 | 5.78 |
| 1973 | 1112 | 469 | 42.18 | 44.97 | 127.22 | 5.40 | 5.26 |
| 1974 | 1309 | 536 | 40.95 | 50.92 | 143.02 | 6.02 | 5.84 |
| 1975 | 1533 | 560 | 36.53 | 52.74 | 148.67 | 6.21 | 6.02 |
| 1976 | 1419 | 524 | 36.93 | 48.92 | 136.96 | 5.81 | 5.65 |
| 1977 | 1468 | 529 | 36.04 | 49.00 | 140.24 | 5.68 | 5.52 |
| 1978 | 1523 | 555 | 36.44 | 51.07 | 143.3 | 5.88 | 5.71 |
| 1979 | 1557 | 555 | 35.65 | 50.84 | 134.29 | 5.55 | 5.40 |
| 1980 | 1517 | 529 | 34.87 | 48.30 | 122.15 | 5.26 | 5.13 |
| 1981 | 1530 | 510 | 33.33 | 46.38 | 119.84 | 4.99 | 4.86 |
| 1982 | 1561 | 528 | 33.82 | 47.72 | 117.56 | 5.14 | 5.01 |
| 1983 | 1615 | 603 | 37.34 | 54.18 | 127.46 | 5.44 | 5.30 |
| 1984 | 1712 | 643 | 37.56 | 57.56 | 137.02 | 5.82 | 5.65 |
| 1985 | 1752 | 634 | 36.19 | 56.60 | 131.84 | 5.85 | 5.68 |
| 1986 | 1731 | 599 | 34.60 | 53.32 | 121.74 | 5.05 | 4.93 |
| 1987 | 1680 | 607 | 36.13 | 53.75 | 125.13 | 5.18 | 5.05 |
| 1988 | 1718 | 569 | 33.12 | 50.02 | 113.80 | 4.63 | 4.53 |
| 1989 | 1740 | 536 | 30.80 | 46.77 | 100.11 | 4.16 | 4.07 |
| 1990 | 1973 | 619 | 31.37 | 53.62 | 115.51 | 4.75 | 4.64 |
| 1991 | 1963 | 622 | 31.69 | 53.59 | 110.56 | 4.69 | 4.58 |
| 1992 | 1976 | 599 | 30.31 | 51.51 | 111.16 | 4.46 | 4.36 |
| 1993 | 2034 | 680 | 33.43 | 58.43 | 116.94 | 4.91 | 4.79 |
| 1994 | 2182 | 792 | 36.30 | 68.02 | 127.76 | 5.81 | 5.64 |
| 1995 | 2313 | 804 | 34.76 | 69.07 | 127.65 | 5.46 | 5.31 |
| 1996 | 2231 | 794 | 35.59 | 68.13 | 118.33 | 5.18 | 5.05 |
| 1997 | 2064 | 802 | 38.86 | 68.71 | 114.69 | 5.32 | 5.18 |

**Table 1** (*continued*)

| Year | No. of cancer death | No. of liver cancer death | Of total cancer (%) | Liver cancer mortality rate (per $10^5$) | Truncated rate of 35–64 (per $10^5$) | Cumulative rate of 0–74 (%) | Cumulative risk (%) |
|---|---|---|---|---|---|---|---|
| 1998 | 2269 | 803 | 35.39 | 68.83 | 111.38 | 5.09 | 4.96 |
| 1999 | 2266 | 797 | 35.17 | 68.51 | 118.64 | 5.04 | 4.91 |
| 2000 | 2407 | 810 | 33.65 | 69.73 | 114.43 | 5.05 | 4.92 |
| 2001 | 2347 | 853 | 36.34 | 73.58 | 115.23 | 5.04 | 4.92 |
| 2002 | 2399 | 863 | 35.97 | 74.79 | 115.67 | 5.05 | 4.93 |
| 2003 | 2456 | 809 | 32.94 | 70.50 | 104.53 | 4.67 | 4.57 |
| 2004 | 2656 | 913 | 34.38 | 80.16 | 110.07 | 5.43 | 5.29 |
| 2005 | 2897 | 863 | 29.79 | 76.28 | 102.88 | 4.93 | 4.81 |
| 2006 | 2818 | 841 | 29.84 | 74.65 | 96.44 | 4.89 | 4.77 |
| 2007 | 2853 | 802 | 28.11 | 71.52 | 91.19 | 4.24 | 4.15 |
| 2008 | 3230 | 885 | 27.40 | 79.28 | 94.67 | 4.85 | 4.73 |
| 2009 | 3043 | 798 | 26.22 | 71.57 | 80.65 | 4.26 | 4.17 |
| 2010 | 3163 | 751 | 23.74 | 67.16 | 80.29 | 3.70 | 3.64 |
| 2011 | 3324 | 746 | 22.44 | 66.47 | 75.38 | 3.81 | 3.74 |
| 2012 | 3127 | 745 | 23.82 | 66.28 | 73.05 | 3.84 | 3.77 |
| 2013 | 3199 | 703 | 21.98 | 62.56 | 63.85 | 3.51 | 3.44 |
| 2014 | 3142 | 649 | 20.66 | 57.77 | 55.09 | 3.02 | 2.98 |
| 2015 | 3451 | 686 | 19.88 | 61.15 | 58.70 | 3.41 | 3.36 |
| 2016 | 3494 | 651 | 18.63 | 58.12 | 49.91 | 3.09 | 3.04 |
| 2017 | 3523 | 588 | 16.69 | 52.61 | 43.92 | 2.84 | 2.80 |
| **Total\*** | **102390** | **31261** | **30.53** | **60.48** | **104.06** | **4.79** | **4.68** |

**Notes.**
*Sum of 1972–2017 registered data.

showed that the AAPCs for men, women and both sexes were −1.8% (95% CI [−2.2% to −1.5%]), −1.4% (95% CI [−3.5%–0.7%]) and −1.8% (95% CI [−2.8% to −0.8%]), respectively, as shown in Table 2.

## Crude mortality, standardized mortality of liver cancer by gender

The mortality rates of liver cancer over time in Qidong have been relatively stable; gender differences always higher in males than in females. From 1972 to 2007, the crude mortality rate of liver cancer was $91.86/10^5$ in men and $29.92/10^5$ in women, with a sex ratio of 3.07:1. For ASRC and ASRW, the sex ratio was 3.48:1 and 3.32:1, respectively. The crude mortality rates by sex and by year are listed in Table 3.

## Age specific mortality and mean age of death

The mortality rate of liver cancer increased with age. At the age group of 35–39, the rate ($67.59/10^5$) exceeded the average mortality level ($58.86/10^5$) in the population. Mortality rates at all age groups over 40 remained at levels of around $110/10^5$, and dropped to less than $90/10^5$ at the age group of 80–84. Rates in all age groups were also higher in men than in women. From 1972 to 2017, the average age of death of liver cancer in Qidong was 53.71 years old, but showing a 5-year difference by gender: men 52.41 and women 57.58. It was observed that the average age of death for both sexes was under 50 years each year before

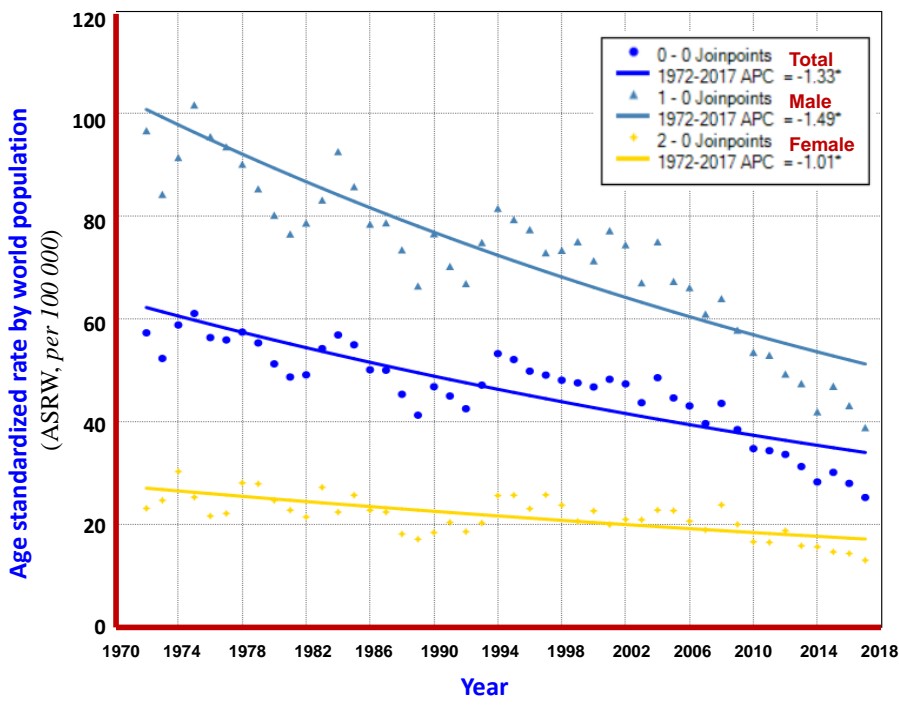

**Figure 2** **AAPC models of ASRW by sex for liver cancer mortality in Qidong, 1972–2017.** Each point represents the actual death rate of liver cancer for males (▲), females (♦) and both sexes (●). The Joinpoint Regression analysis shows that AAPCs of ASRW were −1.49% for men (━), −1.01% for women (━), and −1.33% for both sexes (━).

1993 (reaching a low of 48); since 1993, age of death of liver cancer has increased rapidly from 50, to 55 in 2004, and then over 60 after 2012 (Fig. 4).

## Mortality trend of liver cancer by period

Taking 1972 as the first period, every 5 years as the next increments, 9 more periods covered the following 45 years in order to analyze the mortality rate of each age group. The results show that the mortality rate of liver cancer in the age group ≥ 75 years fluctuated in the range of 50–70 per $10^5$ before the 1990s, and then gradually increased to more than $100/10^5$, and by 2013–2017, it had exceeded $200/10^5$. At age groups of 55–64 and 65–74 the rates have swayed around the range of 100–150 per $10^5$. A slight upward trend in the later period was found in the 55–64 age group, while a slight downward trend was seen at 45–54 age group. There were significant declining trends in all age groups under 54 years old, particularly at 35–44, 25–34, and 15–24 (Fig. 5).

The analysis with the Joinpoint Regression Program showed that the mortality rate at age group ≥ 75 years had a significant upwards trend in 1972–2017, with an AAPC of +3.69%; and at age group of 65–74 years, AAPC was +0.95%. The younger the age, the more significant the trend for declining mortality from liver cancer: AAPCs at 55–64, 45–54, and 35–44 were −0.85%, −1.73% and −3.58%, respectively (Fig. 6).

**Table 2** The APC and AAPC of liver cancer mortality by multiple models simultaneously in Qidong, 1972–2017.

| Cohort | Trend (Segment) | Period | APC | 95% CI | Test statistic (t) | Prob >\|t\| |
|---|---|---|---|---|---|---|
| *APC:* | | | | | | |
| Both sexes | 1 | 1972–1992 | −1.4* | −1.8 to −0.9 | −6.1 | 0.00 |
| | 2 | 1992–1995 | 4.5 | −9.9 to 21.2 | 0.6 | 0.50 |
| | 3 | 1995–2006 | −1.4* | −2.5 to −0.3 | −2.7 | 0.00 |
| | 4 | 2006–2017 | −4.7* | −5.7 to −3.7 | −9.3 | 0.00 |
| Male | 1 | 1972–2005 | −0.8* | −1.1 to −0.6 | −7.0 | 0.00 |
| | 2 | 2005–2017 | −4.5* | −5.6 to −3.4 | −8.4 | 0.00 |
| Female | 1 | 1972–1986 | −0.4 | −1.7 to 0.9 | −0.7 | 0.50 |
| | 2 | 1986–1989 | −11.5 | −33 to 16.9 | −0.9 | 0.40 |
| | 3 | 1989–1994 | 7.7 | −1.1 to 17.2 | 1.8 | 0.10 |
| | 4 | 1994–2008 | −1.1 | −2.3 to 0.0 | −2.0 | 0.10 |
| | 5 | 2008–2017 | −4.7* | −6.8 to −2.6 | −4.5 | 0.00 |
| *AAPC:* | | | | | | |
| Both sexes | All period 1972–2017 | | −1.3* | −1.6 to −1.1 | −10.2 | 0.00 |
| | 3 Joinpoints | | −1.8* | −2.8 to −0.8 | −3.5 | 0.00 |
| Male | All period 1972–2017 | | −1.5* | −1.7 to −1.2 | −11.7 | 0.00 |
| | 1 Joinpoint | | −1.8* | −2.2 to −1.5 | −10.8 | 0.00 |
| Female | All period 1972–2017 | | −1.0* | −1.3 to −0.7 | −6.4 | 0.00 |
| | 4 Joinpoints | | −1.4 | −3.5 to 0.7 | −1.3 | 0.20 |

**Notes.**
*Indicate that it is significantly different from zero at the alpha = 0.05.

**Liver cancer mortality in the birth cohorts**

Birth cohort analysis demonstrated that the mortality of liver cancer in age groups of 15–19, 20–24, 25–29, 30–34, 35–39 and 40–44 displayed downward trends. At age groups of 45–49 and 50–54 the mortality began to show a downward trend in the later period of the birth cohort. In general, in the cohorts of "1953" and later births, clear cohort effects have been observed with respect to reduction of liver cancer mortality. However, in the cohort of before "1938", i.e., at age groups of 70–74, 75–79, 80 and over, the mortality rates showed rising trends (Fig. 7).

## DISCUSSION

The premise of cancer control is to understand the prevalence and trends of cancer. Therefore, long-run monitoring data on incidence or mortality are particularly of importance. Since the 1970s, Qidong City, Jiangsu Province has carried out a series of cancer research programs on prevention and treatment, such as retrospective survey of death, cancer registration reporting, studies of cancer etiology and prevention (*Zhu, Chen & Huang, 1980*; *Chen et al., 2019*; *Chen, 2013*) that provide detailed information for the evaluation of long-term trends and comprehensive prevention effects of liver cancer mortality in Qidong, and in China as well.

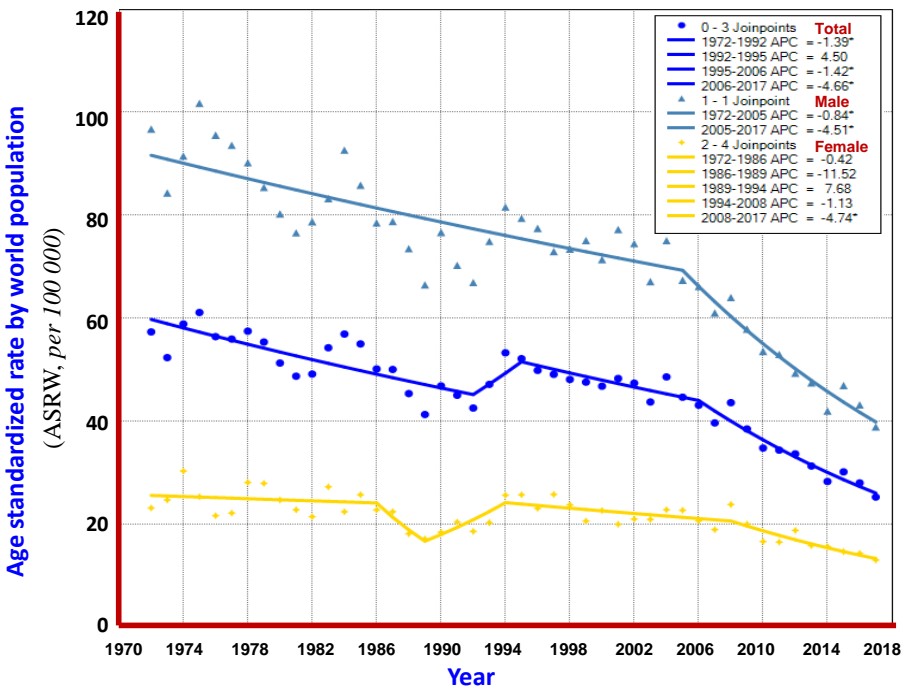

**Figure 3  The APC of liver cancer mortality in Qidong, 1972–2017.** Each point represents the actual death rate of liver cancer for males (▲), females (✦) and both sexes (●). Turning point analysis of multiple models shows that the trend in ASRW of liver cancer during the period could be divided into three join-points (four periods) for both sexes (━━). For men, it was one point (2 periods, ━━), and for women, four joinpoints (5 periods, ━━).

Data over the past 60 years (Fig. 1) show that the natural death rate in the Qidong population experienced an evolutive course from higher to lower, and then from lower back to higher, truly reflecting the dynamic characteristics of the local population due to substantial change of population structure (*Chen, 2013*; *Chen et al., 2013b*) and control of diseases: the slow decline of mortality rate (such as infectious diseases) in the general population in the early period, and the increase of mortality rate (such as chronic diseases) due to aging in the recent decades. That the trend of cancer mortality has increased likely reflects the synergetic effects of two factors: an increase of risk factors and the aging of the population. It is worth noting that the trend of liver cancer mortality before the beginning of this century was "synchronous" with the rate of increase of cancer; after 2008, liver cancer mortality was no longer elevated in concert with the increase of overall cancer mortality, but rather decreased. Other cancers such as lung cancer are now exerting dominance in Qidong.

Our data shows that liver cancer has been the primary form of cancer in Qidong since 1958. The observed crude mortality rate of liver cancer increased from $20.45/10^5$ in the 1950's to about $50/10^5$ in the middle 1970's, and reached to the highest level of more than $80/10^5$ at the turn of this century. Moreover, the proportion of liver cancer in all cancers was 35.49% (4428/12476) in 1958-1971 and 30.53% (31261/102390) in 1972–2017, meaning one in three cancer patients died of liver cancer. However, the proportion of liver cancer

**Table 3** Crude rate, ASRC, and ASRW by sex for liver cancer mortality in Qidong, 1972–2017 (per $10^5$).

| Year | Male | | | | Female | | | | Total | | | |
|------|------|------|------|------|------|------|------|------|------|------|------|------|
| | No. of Death | CR | ASRC | ASRW | No. of Death | CR | ASRC | ASRW | No. of Death | CR | ASRC | ASRW |
| 1972 | 394 | 77.76 | 75.96 | 96.73 | 113 | 21.43 | 18.01 | 23.16 | 507 | 49.04 | 45.11 | 57.34 |
| 1973 | 352 | 68.85 | 66.70 | 84.32 | 117 | 22.01 | 20.30 | 24.71 | 469 | 44.97 | 41.97 | 52.36 |
| 1974 | 390 | 75.43 | 73.34 | 91.49 | 146 | 27.26 | 23.62 | 30.33 | 536 | 50.92 | 47.02 | 58.87 |
| 1975 | 432 | 82.71 | 80.99 | 101.73 | 128 | 23.72 | 19.10 | 25.33 | 560 | 52.74 | 48.25 | 61.12 |
| 1976 | 416 | 78.88 | 77.61 | 95.54 | 108 | 19.87 | 16.98 | 21.66 | 524 | 48.92 | 45.67 | 56.42 |
| 1977 | 415 | 78.05 | 74.07 | 93.58 | 114 | 20.81 | 16.90 | 22.17 | 529 | 49.00 | 44.06 | 55.96 |
| 1978 | 415 | 77.54 | 71.85 | 90.19 | 140 | 25.38 | 22.25 | 28.11 | 555 | 51.07 | 45.93 | 57.49 |
| 1979 | 409 | 76.06 | 68.78 | 85.39 | 146 | 26.35 | 22.21 | 27.93 | 555 | 50.84 | 44.61 | 55.38 |
| 1980 | 395 | 73.42 | 63.65 | 80.27 | 134 | 24.05 | 19.29 | 24.71 | 529 | 48.30 | 40.67 | 51.30 |
| 1981 | 388 | 71.81 | 61.55 | 76.59 | 122 | 21.81 | 17.78 | 22.80 | 510 | 46.38 | 39.08 | 48.75 |
| 1982 | 411 | 75.31 | 62.37 | 78.76 | 117 | 20.87 | 16.75 | 21.47 | 528 | 47.72 | 39.01 | 49.16 |
| 1983 | 448 | 81.59 | 66.26 | 83.21 | 155 | 27.49 | 21.11 | 27.26 | 603 | 54.18 | 43.20 | 54.27 |
| 1984 | 511 | 92.69 | 74.72 | 92.64 | 132 | 23.33 | 17.41 | 22.43 | 643 | 57.56 | 45.65 | 56.93 |
| 1985 | 480 | 86.66 | 67.39 | 85.81 | 154 | 27.19 | 19.92 | 25.73 | 634 | 56.60 | 43.25 | 55.02 |
| 1986 | 459 | 82.59 | 63.28 | 78.51 | 140 | 24.66 | 17.67 | 22.79 | 599 | 53.32 | 40.15 | 50.15 |
| 1987 | 467 | 83.63 | 63.16 | 78.78 | 140 | 24.52 | 17.44 | 22.44 | 607 | 53.75 | 39.99 | 50.06 |
| 1988 | 454 | 80.64 | 59.55 | 73.50 | 115 | 20.02 | 14.32 | 18.17 | 569 | 50.02 | 36.67 | 45.37 |
| 1989 | 421 | 74.18 | 53.47 | 66.48 | 115 | 19.88 | 13.54 | 17.15 | 536 | 46.77 | 33.26 | 41.31 |
| 1990 | 495 | 86.52 | 60.88 | 76.67 | 124 | 21.29 | 14.97 | 18.44 | 619 | 53.62 | 37.55 | 46.85 |
| 1991 | 475 | 82.51 | 56.14 | 70.28 | 147 | 25.13 | 15.31 | 20.42 | 622 | 53.59 | 35.54 | 45.04 |
| 1992 | 465 | 80.54 | 53.86 | 66.95 | 134 | 22.88 | 14.63 | 18.61 | 599 | 51.51 | 34.11 | 42.56 |
| 1993 | 529 | 91.46 | 57.75 | 74.93 | 151 | 25.79 | 14.78 | 20.30 | 680 | 58.43 | 36.06 | 47.16 |
| 1994 | 594 | 102.79 | 62.82 | 81.58 | 198 | 33.77 | 19.12 | 25.64 | 792 | 68.02 | 40.78 | 53.29 |
| 1995 | 600 | 104.01 | 62.01 | 79.41 | 204 | 34.74 | 18.88 | 25.72 | 804 | 69.07 | 40.21 | 52.14 |
| 1996 | 600 | 103.78 | 59.01 | 77.45 | 194 | 33.04 | 17.24 | 23.09 | 794 | 68.13 | 37.93 | 49.88 |
| 1997 | 584 | 100.82 | 56.17 | 72.96 | 218 | 37.07 | 18.40 | 25.79 | 802 | 68.71 | 37.13 | 49.11 |
| 1998 | 600 | 103.77 | 56.67 | 73.41 | 203 | 34.49 | 17.38 | 23.77 | 803 | 68.83 | 36.77 | 48.12 |
| 1999 | 619 | 107.41 | 57.68 | 75.10 | 178 | 30.32 | 15.41 | 20.62 | 797 | 68.51 | 36.42 | 47.59 |
| 2000 | 608 | 105.58 | 54.24 | 71.40 | 202 | 34.49 | 16.46 | 22.66 | 810 | 69.73 | 35.22 | 46.80 |
| 2001 | 667 | 116.12 | 59.61 | 77.24 | 186 | 31.80 | 14.31 | 19.99 | 853 | 73.58 | 36.76 | 48.30 |
| 2002 | 669 | 117.00 | 56.22 | 74.49 | 194 | 33.32 | 15.04 | 21.01 | 863 | 74.79 | 35.46 | 47.40 |
| 2003 | 605 | 106.42 | 50.16 | 67.10 | 204 | 35.23 | 15.00 | 20.94 | 809 | 70.50 | 32.45 | 43.74 |
| 2004 | 684 | 121.53 | 56.17 | 75.08 | 229 | 39.75 | 15.90 | 22.82 | 913 | 80.16 | 35.87 | 48.61 |
| 2005 | 631 | 112.98 | 49.42 | 67.38 | 232 | 40.50 | 15.63 | 22.71 | 863 | 76.28 | 32.36 | 44.66 |
| 2006 | 619 | 111.35 | 48.62 | 66.16 | 222 | 38.90 | 13.95 | 20.67 | 841 | 74.65 | 31.14 | 43.11 |
| 2007 | 590 | 106.68 | 44.41 | 61.02 | 212 | 37.30 | 13.00 | 18.95 | 802 | 71.52 | 28.58 | 39.67 |
| 2008 | 624 | 113.41 | 46.94 | 64.01 | 261 | 46.11 | 16.19 | 23.83 | 885 | 79.28 | 31.44 | 43.61 |
| 2009 | 581 | 105.87 | 41.18 | 57.89 | 217 | 38.33 | 13.75 | 20.03 | 798 | 71.57 | 27.27 | 38.50 |
| 2010 | 549 | 99.86 | 38.83 | 53.57 | 202 | 35.54 | 11.04 | 16.64 | 751 | 67.16 | 24.81 | 34.79 |

**Table 3** (*continued*)

| Year | Male | | | | Female | | | | Total | | | |
|---|---|---|---|---|---|---|---|---|---|---|---|---|
| | No. of Death | CR | ASRC | ASRW | No. of Death | CR | ASRC | ASRW | No. of Death | CR | ASRC | ASRW |
| 2011 | 545 | 98.84 | 37.76 | 52.98 | 201 | 35.21 | 10.91 | 16.54 | 746 | 66.47 | 24.16 | 34.39 |
| 2012 | 522 | 94.54 | 34.00 | 49.31 | 223 | 39.00 | 12.57 | 18.81 | 745 | 66.28 | 23.10 | 33.66 |
| 2013 | 496 | 89.93 | 32.53 | 47.47 | 207 | 36.18 | 9.78 | 15.86 | 703 | 62.56 | 20.99 | 31.30 |
| 2014 | 451 | 81.91 | 28.01 | 41.99 | 198 | 34.57 | 9.81 | 15.66 | 649 | 57.77 | 18.69 | 28.30 |
| 2015 | 494 | 89.89 | 30.76 | 46.94 | 192 | 33.55 | 9.17 | 14.68 | 686 | 61.15 | 19.69 | 30.17 |
| 2016 | 464 | 84.60 | 27.70 | 43.19 | 187 | 32.71 | 8.68 | 14.36 | 651 | 58.12 | 17.87 | 28.01 |
| 2017 | 411 | 75.13 | 24.45 | 38.88 | 177 | 31.02 | 7.86 | 13.04 | 588 | 52.61 | 15.87 | 25.27 |
| **Total** | **23428** | **91.86** | **54.55** | **71.18** | **7833** | **29.92** | **15.67** | **21.44** | **31261** | **60.48** | **34.78** | **45.71** |

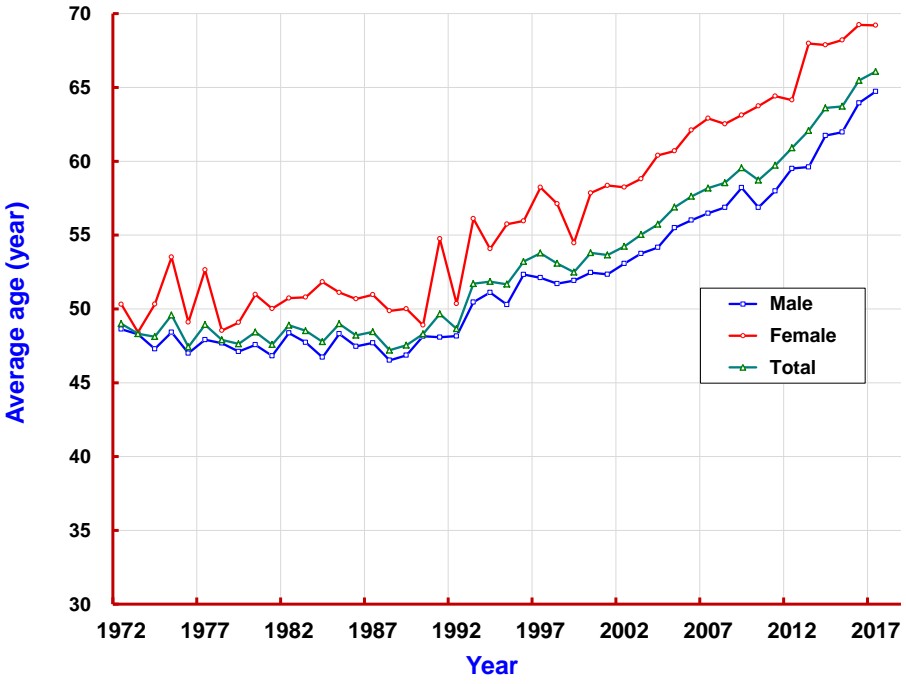

**Figure 4** **The average ages of patients with liver cancer in Qidong, 1972–2017.** The mortality rate of liver cancer increased with age. The average age of death from liver cancer in Qidong for both sexes (—▲—), men (—□—), and women (—○—).

in total cancer death has declined substantively over the past decade. Table 1 shows that the proportion of liver cancer in total cancer deaths has declined to 22.52% (8004/35549) in the years 2008–2017, and to 18.39% (1925/10468) in just the last three years of analysis (2015–2017).

Based upon the data from the registration system, the overall mortality trend of liver cancer was evaluated. It can be seen that the percent change (PC) of the crude mortality of liver cancer during 1972–2017 was +17.78%; but the PCs of ASRC and ASRW were −61.25% and −51.43%, showing a downward trend. Joinpoint Regression analysis

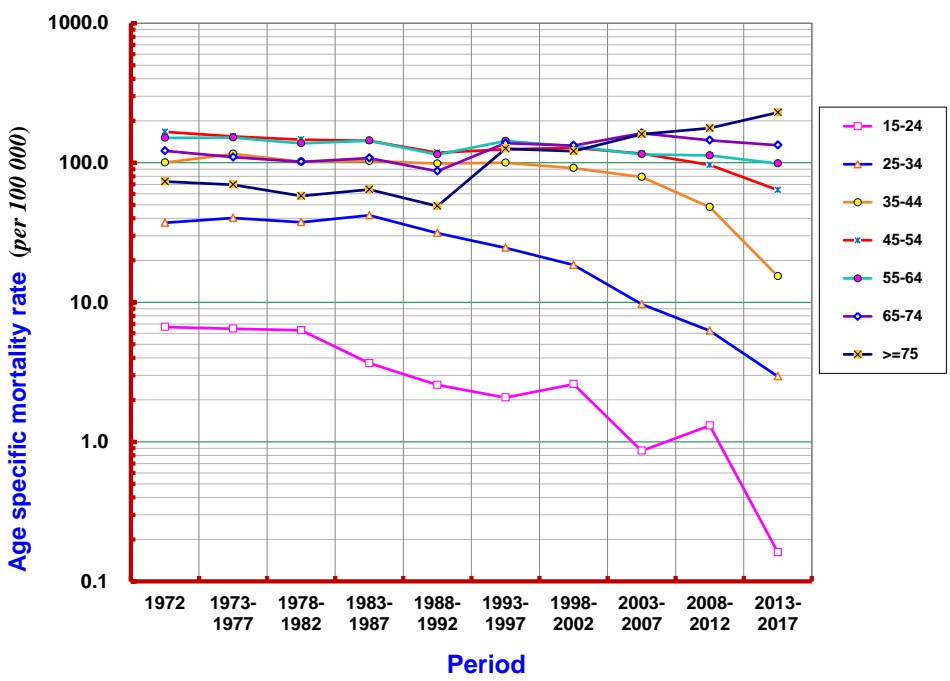

**Figure 5 Age-specific mortality rates of liver cancer by period in Qidong, 1972–2017.** There were significant declining trends for liver cancer mortality rates in all age groups under 54 years old, particularly at age groups of 35–44, 25–34, and 15–24. A slight upward trend in the later period was observed at 55–64, while a slight downward trend was seen at 45–54.

illustrated that average annual percent change (AAPC), in term of ASRW, displayed a clear declining trend, which was −1.49% in men, −1.01% in women, and −1.33% for both sexes (Fig. 2). According to the multiple model of Joinpoint Regression (Table 2), the AAPCs of men, women and the both were −1.8%, −1.4% and −1.8%, respectively, indicating that mortality from liver cancer in Qidong did show a significant downward trend.

Similar trends have been found across China: the standardized mortality rate of liver cancer, for instance, in Changning District, Shanghai during the years 1973-2013, was −2.1% in men and −2.7% in women, respectively (*Ji et al., 2020*). A recent study in China showed that the crude mortality rate of liver cancer in China increased during 1997-2016, with an AAPC of +0.5%, but the age-standardized rate was in decline (*Ding et al., 2019*). Another report showed that the age-standardized mortality rate of AAPC for liver cancer in China was −0.5% in men, −1.3% in women, and -0.8% in both sexes from 1990 to 2017 (*Wang et al., 2019*). A South Korea report showed that the age-standardized mortality rate of AAPC in 1983–2012 was −1.55% in men and −0.56% in women (*Lim, Ha & Song, 2015*). In a recent Meta-analysis report (*Dasgupta et al., 2020*) that included 31 population-based studies showed that the pooled APC estimates for liver cancers was +0.8%, in which +3.2% in the region of North America/Europe/Australia, whereas −1.7% in several Asian countries. These different patterns may reflect geographical variation or changes in the ratio of liver cancers attributable to various risk factors.

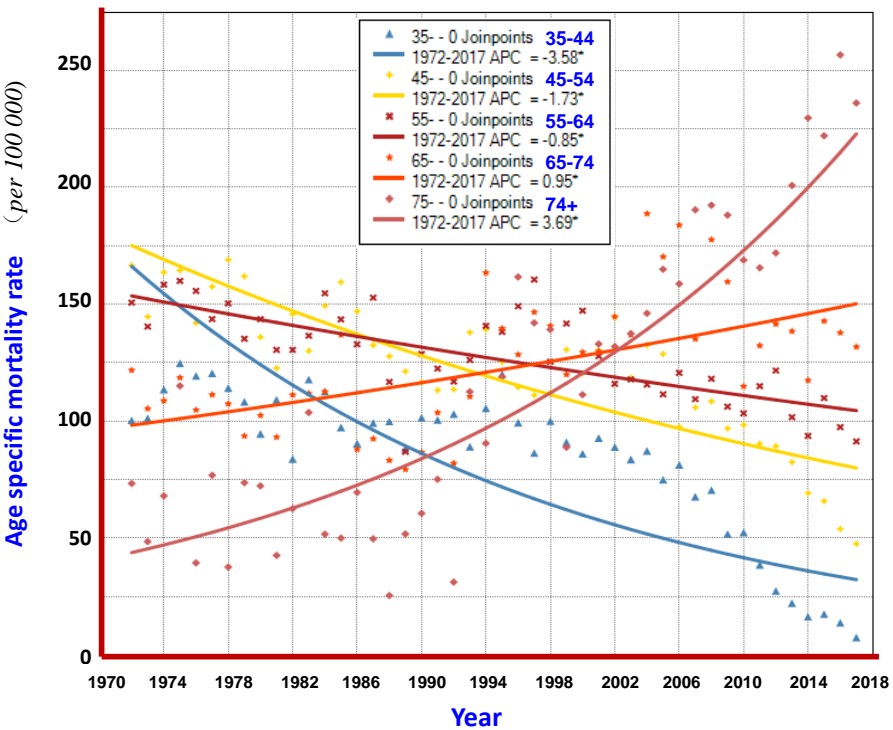

**Figure 6** **AAPCs of mortality rates of liver cancer by age group in Qidong, 1972–2017.** The Joinpoint Regression analysis shows that the mortality rates at age group of 75+ years, and 65–74 years had significant upwards trends, with AAPC of +3.69%, and +0.95%, respectively. The AAPCs at younger age groups of 55–64, 45–54, and 35–44 had declining trends, being −0.85%, −1.73% and −3.58%, respectively.

The age-specific mortality rate from this 60-year data set showed that liver cancer contributed the most hazard towards disease burden in the middle-aged and elderly residents. In individuals over 35 years in Qidong, the mortality rate of liver cancer (over $110/10^5$) has exceeded the average level of the general population, and has been stable at the level of $110–140/10^5$ for those over 45 years old. The cumulative rate and the cumulative risk of 0–74 years old (which reflects the level of lifetime mortality) reached 5.15% and 5.02%, respectively, being yet another metric of this cancer epidemic. In the 1970s and 1980s, it was found that the proportion of liver cancer deaths in young adults was very high, in term of the truncated rate of 35–64 years: between $120/10^5$ and $140/10^5$ in the 1970s–1980s, down to $100–110/10^5$ in 1990–2005, to less than $100/10^5$ in the last period, and even lower than $50/10^5$ in the most recent 2 years. For the average death age of liver cancer, they were relatively young patients before the 1990s, since then the average death age increased gradually from under 50 to exceed 65 years old. These changes clearly show that the mortality of liver cancer in the Qidong youth and middle-aged population has exhibited a rapid downward trend.

Furthermore, the change of liver cancer mortality in each age group by 5-year-period was investigated (Fig. 5). Gradual downward trends could be found in the age groups of 15–24, 25–34, and 35–44 years old; but obvious up- or down-ward trends were not observed in

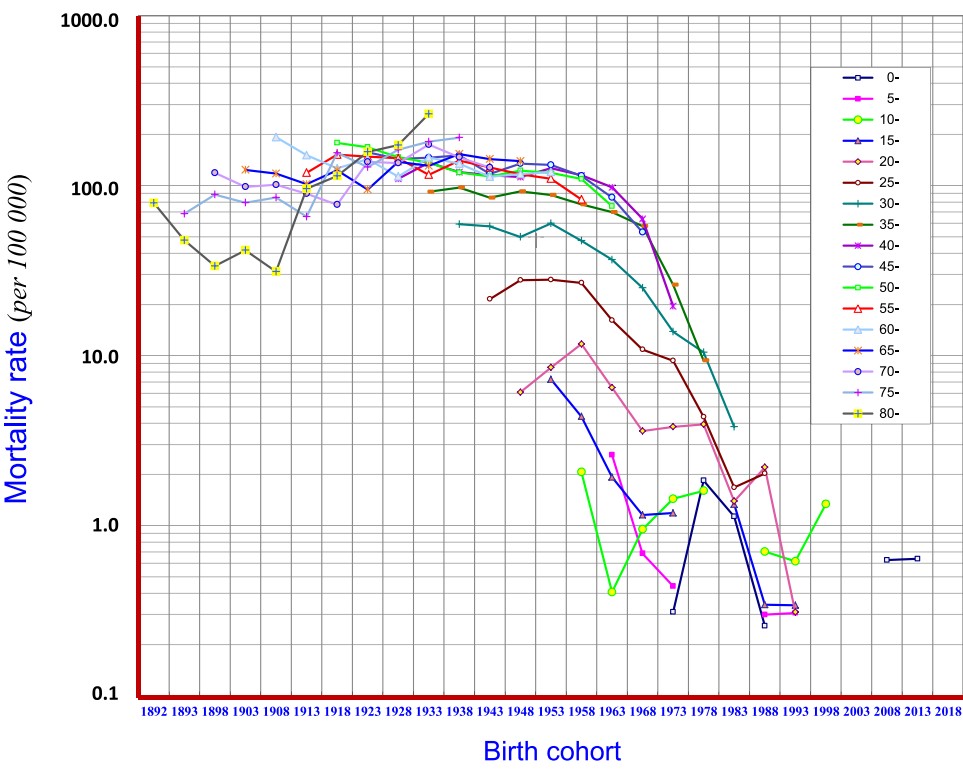

**Figure 7 Birth-cohort mortality rates of liver cancer in Qidong, 1972–2017.** Birth cohort analysis shows that the mortality of liver cancer in age groups of 15–19, 20–24, 25–29, 30–34, 35–34 and 40–44 displayed downward trends (in the cohorts of "1953" and later births), but showed rising trends in age groups of 70–74, 75–79, 80 and over (in the cohorts before "1938"). Mortality began to show a downward trend in the later periods of the birth cohorts in ages of 45–49, and 50–54

the age groups of 45–54 and over, whilst fluctuated at high levels of mortality. However, the mortality of liver cancer in the age group over 75 years has continuously increased so far. Joinpoint regression analysis also shows that the AAPCs in those aged 35–44, 45–54 and 55–64 exhibited significantly decreasing trends, while in those aged 65–74, 75 and over exhibited significantly increasing trends (Fig. 6). These results demonstrate that the high mortality rate for liver cancer in the elderly population, together with the increased proportion of the elderly population (aging), has had a great impact on the increasing crude mortality rate of liver cancer in this area. In the United States, the largest increases were observed in persons aged 55–59 years (AAPC, 8.9%) and 60–64 years (AAPC, 6.4%) between 2000 and 2012 (*White et al., 2017*).

The encouraging elements of this situation are, however, based upon the birth-cohort analysis. The cohort mortality rates in all age groups of 15–19, 20–24, 25–29, 30–34, 35–39 and 40–44 years old, i.e., for those born after the 1950s, have experienced weakened epidemic trends (Fig. 7). This outcome has given a clear signal towards the prevention and control of liver cancer: the reduction of liver cancer mortality in the Qidong population has already been reflected in the young generation of this area; and, this downward trend is also being transmitted to the middle-aged cohort population. Thus, the key question is:
what kind of causes (measures) led to the change (decline) of liver cancer mortality in the young and middle-aged people in Qidong?

Globally, there are many of risk factors contributing to liver cancer etiology, and different populations have disparate risk patterns (*Pham et al., 2018*). Primary prevention precisely aimed at key risk factors is essential and perhaps the only realistic and sustainable approach (*Yang et al., 2019*). During the past decades in the Qidong region, epidemiological and etiological research on liver cancer, control of risk factors, implementation of primary prevention (elimination of exposure to risk factors) and secondary prevention (early diagnosis and early treatment) have been the priority goals for liver cancer studies (*Zhu, Chen & Huang, 1980*; *Chen, 2013*; *Chen et al., 2005*; *Chen et al., 2014*). This effort has achieved positive results. The main measures that have been targeted to the major causes of liver cancer include: prevention of hepatitis B, prevention of grain-mildew leading to aflatoxin contamination and detoxification, improvement of drinking water source and secondary prevention. Which of these measures has made the most direct and/or significant contributions to the decrease of liver cancer mortality in Qidong? It is worth assessing the available information and evidence.

Hepatitis B virus (HBV) is known as perhaps the most important pathogenic factor of liver cancer etiology. Experimental research on HBV, and control of hepatic diseases have been undertaken since the 1970s in Qidong (*Zhu, Chen & Huang, 1980*; *Chen et al., 2014*; *Chen et al., 2010*; *Sun et al., 2002*). Most important, hepatitis B vaccination in infants started in 1983: the newborns in some townships would receive three doses of hepatitis B vaccine, but the limited coverage of this program led to only a small proportion of actual vaccinations in the newborns prior to 2002. Then, hepatitis B vaccination was included into the Expanded Program on Immunization (EPI) in China (*Hutton, So & Brandeau, 2010*; *Luo, Li & Ruan, 2012*) which made it possible to vaccinate all newborns with full population (and financial) coverage in Qidong (*Sun et al., 2002*). It does mean though, that as of the year 2017, most of the hepatitis B vaccine recipients have been under the age of 20 years, who in turn did not contribute to the real changes observed in the mortality curve. In other words, it is obvious that the curve on the right side of the ''1983'' birth cohort as in Fig. 7 is not enough to affect the downtrend of mortality rate of liver cancer in the Qidong population. But, if one focuses on the period results in Fig. 4 one would speculate that the decline curve in the age group of 15–24, for instance, is likely due to vaccination. Obviously, the cross-sectional results (Fig. 5) obscure the real effects within the cohort results (Fig. 7). An early report from Taiwan proffered a conclusion that the incidence of hepatocellular carcinoma in children has declined due to the hepatitis B vaccination (*Chang et al., 1997*). But, we have noted that even in the figure of their paper, the major decline of incidence for the children of 6–14 years in fact occurred before 1984 when the vaccination had not yet been introduced. In a later paper (*Chang et al., 2009*) from a 20-year follow up, it was concluded that the prevention of hepatocellular carcinoma by this HBV vaccine extends from childhood to early adulthood. Accordingly, the decrease of liver cancer incidence/mortality in children/adolescence or young adults in Qidong or in Taiwan must be affected by the elimination or diminution of other risk factors (other
than HBV) or other effective measures taken, simply because the changed panorama of liver cancer pattern in the general population happened before vaccination.

The relationship between drinking water and liver cancer in Qidong has also attracted the interest of researchers (*Su, 1979*; *Yu, 1995*). However, it is not clear what kind of substance or pollution in the water was related to the incidence (or mortality) of liver cancer. Possible factors include cyanobacteria toxin (microcystins or cyanotoxins) (*Yu, Zhao & Zi, 2001*; *Lian et al., 2006*; *Labine & Minuk, 2015*), some chemical mutagens (*Ruan, Chen & Zhang, 1997*), or even inorganic arsenic (*Wang, Cheng & Zhang, 2014*) in the water. There have been no reports to provide any strong evidence in animal experiments as to causation of liver cancer, and not enough epidemiological data in the population to establish the relationship between drinking water and the incidence or mortality of liver cancer. It has been reported (*Zhang et al., 2015*) that the distribution of cyanobacteria in the United States is very extensive, but incidence rate of liver cancer has not been high for decades. However, a recent report, based upon an ecological study in Ohio (USA), observed that a population served by bloom-impacted surface waters had 14.2% higher HCC incidence rates than those served by non-bloom-impacted surface waters (*Gorham et al., 2020*). In a Canadian study, however, no association with cyanobacterial toxin exposure in the geographic distribution of liver cancer was observed, implying blue–green algae pollution may not play an important role in the etiology of liver cancer in Canada (*Labine et al., 2015*). Qidong has gradually changed the sources of drinking water since the 1980s and vastly expanded the coverage of piped tap water since the beginning of this century (*Chen, 2003*). Yet the existing evidence related to a role of drinking water is not sufficient to illuminate the inevitable relationship between the decrease of liver cancer mortality and the change of drinking water source and causality in Qidong over the past decades.

There have been many reports on the relationship between aflatoxin and liver cancer, showing that aflatoxin can be a critical etiological factor of liver cancer in some geographical regions (*IARC, 1993*; *Kensler et al., 2011*; *Benkerroum, 2019*). Corn can be easily contaminated by aflatoxin (produced by *Aspergillus flavus*) and was a staple food of Qidong residents for a long time, especially in times of food shortages. In the 1970s, the contamination rate of aflatoxin in corn (levels exceeding 20 ppb) ranged from 5.68–98.79% (*Zhu, Chen & Huang, 1980*). Animal experiments using corn harvested from Qidong fields exhibited definite hepatocarcinogenic effects on rats and ducks in the 1970s (*Zhu, Chen & Huang, 1980*). Since then comprehensive preventive measures to control aflatoxin contamination, so called "post-harvest management", have been undertaken in Qidong (*Zhu, Chen & Huang, 1980*; *Chen et al., 2014*). In response to social and economic changes, from the mid-1980s, Qidong residents (all age groups) completely changed their habit of eating corn to rice (known to be much less contaminated by aflatoxin) (*Zhu, Chen & Huang, 1980*; *Chen et al., 2019*; *Chen et al., 2014*). These changes of habit and other control measures have led to impressive results evidenced by exposure biomarkers detected in historical biobank blood samples: the median levels of aflatoxin-albumin adducts in Qidong residents decreased significantly from 19.3 pg/mg in 1989 to 3.6 in 1995 pg/mg, 2.3 pg/mg in 1999, 1.4 pg/mg in 2003, and undetected (<0.5 pg/mg) levels in 2009 and 2012 (*Chen et al., 2013b*). The exposure level of aflatoxin in this decade has

decreased to a level of >1/40th of the level in the 1980s. To date, this provides the most and perhaps only convincing evidence for the causal link (association) between the elimination of risk factors and the reduction of mortality of liver cancer, especially as supported by the birth-cohort analysis that demonstrates about a 50% decrease of ASRW of liver cancer over the past decades. From Figs. 5 and 7, we also noted that the trend of mortality at aged 75 and over was not decreasing but in fact increasing. This outcome may reflect that the diminution of aflatoxin exposure in the late period of life was too late to affect aging people or may reflect a "delayed toxicity" (*Benkerroum, 2019*). This interpretation is plausible given the long latency period for the development of liver cancer.

There may be other risk factors contributing to liver cancer etiology in Qidong. Hepatitis C virus, for example, is of most interest globally (*Yang et al., 2019*; *Maucort-Boulch et al., 2018*). However, there has been no evidence to show this virus was common in the general population, or in patients with liver cancer in Qidong over the past 60 years. In our previous nested case-control study early this century, we found that the prevalence of HCV infection was low (1.6% in cases *vs.* 2.1% in controls) and did not seem to play a role in the etiology for liver cancer (*Szymañska et al., 2009*). A 13.25 year follow-up study in Qidong also indicated that only 6 of 119 liver cancer patients with positive HBV were coinfected with HCV, contributing little to the risk of hepatocellular carcinoma (*Ming et al., 2002*). Besides the major risk factors as mentioned above, there may be other minor risk factors or the interaction effects of risk factors from environment or from mutagens (*Lian et al., 2006*; *Szymañska et al., 2009*; *Kensler et al., 2003*; *Zhang et al., 2017*). In recent times, diabetes mellitus (DM), metabolic syndrome (MetS) and obesity or metabolic disease/obesity have been considered as growing problems for the development of liver cancer (*Davila et al., 2005*; *Kim & Han, 2012*; *Li, Xu & Gao, 2018*). In a 2015 study from Jiangsu province where Qidong is located, the prevalences of DM were 8.6% in men and 8.4% in women, and 2.7–4.8% at ages of 18–44 and 13.2% at ages over 65 (*Tao et al., 2015*). In a study of 325 patients with liver cancer in our Qidong hospital (*Song et al., 2018*), 31.38% had type II DM, and those who were at age of over 60 *vs.* under 60, the rates were 43.14% *vs.* 13.28%, indicating a possible association between DM and the risk of liver cancer in the elderly. This may also partially explain why the birth cohort of age groups over 65 had increased mortality (Fig. 7). However, in a Taiwan study (*Chen et al., 2013a*), it was concluded that DM, MetS and obesity were not associated with liver cancer development in hepatitis virus infected groups within an epidemic area. These factors need to be further observed and studied. Hence, among these mentioned factors and or the effects on liver cancer, either there was insufficient evidence or a synergistic effect with aflatoxin or HBV synchronously. Furthermore, the population attributable risk (PAR) for developing liver cancer in the general population was limited, so it was unlikely to affect the downward trend of liver cancer mortality in Qidong.

Secondary prevention, which involves early diagnosis and early detection, could influence liver cancer mortality. In the 1970s, a series of large-scale screening programs for early diagnosis and treatment of liver cancer measuring serum alpha-fetoprotein (AFP) were carried out. A large number of early liver cancer patients were detected and some good outcomes achieved (*Zhu, Chen & Huang, 1980*). However, the mortality rates of liver

cancer in Qidong at that period fluctuated at a high level (Fig. 1, Table 1). In a 1980's liver cancer screening program in Qidong, it was found that focused screening within a high-risk population could improve the detection levels of liver cancer, with a short-term effect on the change of survival rate, but no effect on mortality in the long term (*Chen et al., 2003*). After 2007, periodical diagnostic screening using combined methods of AFP and ultrasound examination monitoring were recommended. This practice has shown the effect of screening on the early diagnosis and treatment, and may improve the outcome of liver cancer for some (*Chen et al., 2017*). However, the proportion of patients who received the early diagnosis relative to the total number of patients in Qidong was still too small, and insufficient to affect mortality outcomes in the general population. Unfortunately, in general, the now substantial decrease of liver cancer mortality in Qidong cannot be attributed to a change in the detection rate or to improvements in treatment, but rather to changes in the matrix of risk factors in the population.

## CONCLUSIONS

The crude mortality rate of liver cancer in Qidong experienced a dynamic flux from low to high, then sustained at high levels for decades, followed by a gradual decrease in recent years. After excluding the influence of the ''aging'' of the Qidong population, it emerged that the decrease of the standardized mortality rate of liver cancer in Qidong happened earlier than that of the crude rate, and the decline has accelerated since the beginning of this century. Moreover, it was first reflected in the newest generation, with a higher slope of decline in men who are at 3-fold higher risk than women overall. In a comprehensive investigation of the risk factors that may affect the downward trends in Qidong, the most significant effect appears to be the control of mildew in the staple food (corn), coupled with profound dietary changes effected by social and economic mandates, to result in dramatic declines in the intake of aflatoxin. This change in the exposure was documented directly with the use of analytically rigorous methods for measures of biological markers in archived serum samples collected longitudinally within the population. The beneficial effect of improving drinking water may not be excluded, but direct positive evidence has not been obtained so far. The prevention of hepatic diseases, such as hepatitis B, must not be ignored, since hepatitis B vaccination has shown effectiveness in prevention of liver diseases in children (*Chang et al., 1997*; *Qu et al., 2014*), but has not yet been reflected in the prevention of liver cancer in the adult populations. For certain, the present evidence regarding HBV vaccination is not enough to explain or contribute to the current declining trend of liver cancer mortality in Qidong. The change (reduction) of liver cancer mortality in the past six decades has clearly implied that the efforts in Qidong have shown the promise of the control of liver cancer. Through unremitting efforts in a landscape of social, economic, environmental and medical change, liver cancer in Qidong is no longer the number one cause of cancer mortailty, having been overtaken by lung cancer in the past decade (*Chen, 2013*; *Chen & Kensler, 2014*). The etiological mechanisms need to be explored further, yet the rationale and appropriateness of the measures taken in Qidong area have been proven, and has helped us to better understand the etiology of liver cancer

in this region. Six decades of monitoring the state of liver cancer mortality in Qidong coupled with rigorous, longitudinal monitoring of aflatoxin biomarkers, has provided a working example that reducing aflatoxin contamination can be achieved by interventions at the population level; in our case, has also provided vigorous evidence that controlling aflatoxin may help reduce substantively the burden of morbidity and mortality of liver cancer.

## ACKNOWLEDGEMENTS

We thank our colleagues at the QDLCI, the Qidong People's Hospital, the Shanghai Cancer Institute and Johns Hopkins University who have assisted in our studies. We thank the leadership of the Qidong City government for fostering our collaborations, and most importantly, thank the many residents of Qidong as well as the local doctors for their dedicated participation in these studies. They have been the enablers to probe the etiology and prevention of liver cancer.

### Funding

This work was funded by the US National Institutes of Health through grants R01 CA196610 and R35 CA197222, the Chinese National Key Projects (2008ZX10002-015, 2008ZX10002-017, 2012ZX10002009), the Scientifc Research Projet of ''333 Project'' in Jiangsu (BRA2019030), and the Nantong Science and Technology Project (MS22019008). The funders had no role in study design, data collection and analysis, decision to publish, or preparation of the manuscript.

### Grant Disclosures

The following grant information was disclosed by the authors:
The US National Institutes of Health: R01 CA196610, R35 CA197222.
The Chinese National Key Projects: 2008ZX10002-015, 2008ZX10002-017, 2012ZX10002009.
The Scientifc Research Projet of ''333 Project'' in Jiangsu: BRA2019030.
The Nantong Science and Technology Project: MS22019008.

### Competing Interests

The authors declare there are no competing interests.

### Author Contributions

- Jian-Guo Chen conceived and designed the experiments, performed the experiments, analyzed the data, prepared figures and/or tables, authored or reviewed drafts of the paper, funding, Field administration, and approved the final draft.
- Jian Zhu performed the experiments, analyzed the data, prepared figures and/or tables, authored or reviewed drafts of the paper, and approved the final draft.
- Yong-Hui Zhang, Yong-Sheng Chen, Jian-Hua Lu and Hai-Zhen Chen performed the experiments, prepared figures and/or tables, and approved the final draft.

- Yuan-Rong Zhu performed the experiments, prepared figures and/or tables, field administration, and approved the final draft.
- Ai-Guo Shen performed the experiments, authored or reviewed drafts of the paper, funding, and approved the final draft.
- Gao-Ren Wang performed the experiments, prepared figures and/or tables, administration, and approved the final draft.
- John D. Groopman and Thomas W. Kensler conceived and designed the experiments, performed the experiments, analyzed the data, prepared figures and/or tables, authored or reviewed drafts of the paper, funding, and approved the final draft.

## Data Availability

Raw data are available as a Supplemental File.

## Supplemental Information

Supplemental information for this article can be found online at http://dx.doi.org/10.7717/peerj.10600#supplemental-information.

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
