# Peer review of "Liver cancer mortality over six decades in an epidemic area: what we have learned"

_PeerJ, doi:10.7717/peerj.10600_

## Round 0.1 · original submission · Minor Revisions

The authors mentioned there was rare data to describe decades’ long natural history or changes of a cancer and liver cancer. For the last decades when we compared the survival rates in drug studies including the real-life experiences we could easily find out the differences in cancer mortality of liver cancer. The results are predictable even for an epidemic area.

Abstract section of manuscript must be re-wriitten.

Reviewer 1 ·

Basic reporting

English language is clear and well-written throughout the manuscript. However, there are some minor points that need to be revised:

Line 57: Add reference
Line 261: Revise ‘these different patterns’

Experimental design

Line 152: Why did you select 35-64 as the threshold?
Line 153: What about people that are elder than 74-year-old?

Validity of the findings

Line 226: Please briefly explain the population structure changes over the decades

Additional comments

This study summarized the changes in liver cancer mortality rates over the decades in Qidong. The authors also highlighted the possibility that the change in aflatoxin intake may contribute to the decrease of liver cancer mortality rate.

Reviewer 2 ·

Basic reporting

Professional English throughout
Sufficient literature references
There are many mortality terms in the manuscript and they can be defined more clearly

Experimental design

It is a really original primary research. To tell you the truth, I haven't read an article that investigated such long-term cancer mortality in an epidemic area.
However research question in not well defined and meaningful. Because the result are predictable.

Validity of the findings

Conclusions are well stated and linked to original research question.
However, there is no impact and novelty in the manuscript.

Additional comments

No comment

Reviewer 3 ·

Basic reporting

I appreciate authors for putting great deal of effort into this study that details of Liver Cancer Mortality trends. This study details analyzing data coming from two different sources- one being retrospective death survey data and another being cancer registry data. Study data spans over few decades. Authors have used various metrics such as natural death rate, cancer mortality rate, liver cancer mortality rate in this study to look at trends over many years.

This is well done study with many data points and have certain limitations in terms of presentation.

• As a reviewer, I suggest re writing and rephrasing entire abstract to make more sense. In current structure it is tough to comprehend. Since this manuscript aimed at reaching to wide range of international audience it would be better to have abstract with better sentences.
• As a reviewer when I looked at the abstract and at the rest of the manuscript, there is a clear difference of English between both. I suggest reframing and restructuring sentences in abstract.
• Abstract is repeated twice this manuscript.

• In methods of abstract it should be age standardized rate instead of age stand ardized rate

• In results section of the abstract, the natural death rate percentage symbol should be "%"

• Authors might want to think if this is appropriate way of representing cancer mortality rate in results section of the abstract. I feel it is not in sync with what is detailed in manuscript. In the body of the manuscript it is represented as 10 to the power of 5 but here in abstract it is represented as 105 at five instances.


• At line 99- I understand that ICD-10 system is the latest. Appropriate references to showcase ICD-10 implementation date in China would be helpful. In United States ICD 10 system has not been implemented until later of 2010.

Experimental design

• From line 77-93 in methods I would also suggest how details to include if the data analyzed was deidentified or identified or how the deidentification was done.

• Authors might also add few details about which software was used for data analysis and running the statistics in methods section.

Validity of the findings

no comments

Additional comments

no comments

---

## Round 0.2 · accepted · Accept

The authors have addressed the questions that the reviewers raised in the first edition. The quality of the manuscript has been improved in the current version.

Reviewer 1 ·

Basic reporting

No comment

Experimental design

No comment

Validity of the findings

No comment

Additional comments

The authors have addressed the questions that I raised in the first edition. The quality of the manuscript has been improved in the current version.

Reviewer 3 ·

Basic reporting

no comments

Experimental design

no comments

Validity of the findings

no comments